# GIS-Based Assessment of Hybrid Pumped Hydro Storage as a Potential Solution for the Clean Energy Transition: The Case of the Kardia Lignite Mine, Western Greece

**DOI:** 10.3390/s23020593

**Published:** 2023-01-04

**Authors:** Pavlos Krassakis, Andreas Karavias, Evangelia Zygouri, Christos Roumpos, Georgios Louloudis, Konstantina Pyrgaki, Nikolaos Koukouzas, Thomas Kempka, Dimitris Karapanos

**Affiliations:** 1Centre for Research and Technology, Hellas (CERTH), 52 Egialias St., 151 25 Maroussi, Greece; 2Public Power Corporation of Greece (PPC), Department of Mining Engineering and Closure Planning, 104 32 Athens, Greece; 3GFZ German Research Centre for Geosciences, Telegrafenberg, 14473 Potsdam, Germany; 4Institute of Geosciences, University of Potsdam, Karl-Liebknecht-St. 24–25, 14476 Potsdam, Germany

**Keywords:** hybrid pumped hydro power storage, hydro power, hydro storage, GIS, Kardia mine, AHP, MCDM

## Abstract

Planned decommissioning of coal-fired plants in Europe requires innovative technical and economic strategies to support coal regions on their path towards a climate-resilient future. The repurposing of open pit mines into hybrid pumped hydro power storage (HPHS) of excess energy from the electric grid, and renewable sources will contribute to the EU Green Deal, increase the economic value, stabilize the regional job market and contribute to the EU energy supply security. This study aims to present a preliminary phase of a geospatial workflow used to evaluate land suitability by implementing a multi-criteria decision making (MCDM) technique with an advanced geographic information system (GIS) in the context of an interdisciplinary feasibility study on HPHS in the Kardia lignite open pit mine (Western Macedonia, Greece). The introduced geospatial analysis is based on the utilization of the constraints and ranking criteria within the boundaries of the abandoned mine regarding specific topographic and proximity criteria. The applied criteria were selected from the literature, while for their weights, the experts’ judgement was introduced by implementing the analytic hierarchy process (AHP), in the framework of the ATLANTIS research program. According to the results, seven regions were recognized as suitable, with a potential energy storage capacity from 1.09 to 5.16 GWh. Particularly, the present study’s results reveal that 9.27% (212,884 m^2^) of the area had a very low suitability, 15.83% (363,599 m^2^) had a low suitability, 23.99% (550,998 m^2^) had a moderate suitability, 24.99% (573,813 m^2^) had a high suitability, and 25.92% (595,125 m^2^) had a very high suitability for the construction of the upper reservoir. The proposed semi-automatic geospatial workflow introduces an innovative tool that can be applied to open pit mines globally to identify the optimum design for an HPHS system depending on the existing lower reservoir.

## 1. Introduction

Energy security is one of the basic pillars of any country’s energy policy. Energy storage is a requirement for a reliable national electricity supply. In line with the Paris Agreement [1] and to meet the net zero carbon emissions target by 2050, the necessity for energy storage technologies for long time periods is significant in shaping a decarbonized future. The latest 2021 Glasgow Climate Pact is an important step toward speeding up national climate transition plans in an effort to reduce coal power and greenhouse gas emissions and limiting the average rise in global temperature to 1.5 °C [2].

Renewable energy sources (RES) are already playing a key role in large-scale energy generation and storage. As coal production for electricity generation has gradually decreased in Europe, RES projects (such as solar, wind, and hydropower projects) have increased (Figure 1) in order to satisfy energy demands [3]. Solar and wind energy have proven to be sustainable, efficient, and cost-effective energy generation solutions at large scales [4]; however, sustainable energy storage solutions are equally essential to ensure energy efficiency and security. Hydropower offers electricity production as well as storage technologies, which can be combined with solar and wind into hybrid systems. There are four types of hydropower technologies: run-of-river hydropower, offshore hydropower, storage hydropower, and pumped storage hydropower or pumped hydro storage (PHS) [5]. Currently, Europe has a capacity of 55,055 MW of pumped hydropower [5], with more projects being added every year.

Pumped hydro storage systems are based on the conversion of electric into gravitational energy and vice versa. The basic components of a PHS plant are an upper water and a lower water reservoir. The upper reservoir is constructed near the lower one, with both vertically separated by a considerable height. Excess electricity from RES, such as solar panels and wind turbines, can be utilized to pump water into the upper reservoir via penstocks (water conduits) (Figure 2). The water from the upper reservoir is released into the lower one on demand, resulting in the generation of electricity by means of hydroelectric turbines during high-peak demand periods. Thus, excess energy from renewable sources is stored until it is required [6].

PHS systems are a very effective solution regarding problems caused by fluctuating electricity generation from renewable energy sources due to the rapid release and water pumping [6]. PHS systems have a storage capacity of one month in the lower reservoir [7]. Their cycle efficiency varies between 75–80% [8], and their energy storage efficiency varies between 65–85%. Hybrid pumped hydro storage (HPHS) systems (Figure 2) combine solar and wind energy with hydropower for a more stable electricity generation system [9].

Modern PHS projects usually require the construction of at least one of the water reservoirs; hence, their implementation can be constrained by a number of environmental parameters (topographic and ecologic) [10]. Therefore, abandoned open pit mines and quarries seem to be suitable candidates for future PHS plants, the potential of which have been assessed in a few studies [11]. The closure of many open pit coal mines in the EU following the EU Green Deal requirements offers the opportunity to convert these into HPHS projects [12]. In addition, these systems may contribute to land reclamation of the former mines and ensure public acceptance by avoiding further excavations, while offering an opportunity for efficient re-use and contributing to the rehabilitation of the former mines. Cost advantages are generated from the presence of transport infrastructure and electricity transmission facilities [13]. As expected, the requirements for a successful HPHS system implementation in abandoned open pit mines are strongly related to topography and environmental restrictions (Figure 2).

Open pit mines can be used either as lower or upper reservoirs for the implementation of HPHS technology, depending on their suitability in terms of topography and proximity-based analyses to the natural and manmade features [11]. The implementation of the HPHS technology in former open pit mines involves the full or partial flooding of the open pit and its transformation into a pit lake as a lower reservoir. Additionally, HPHS systems include the construction of an upper reservoir, along with subsurface or surface water conduits and turbines, and require a connection to the adjacent electricity grid [14].

The repurposing of abandoned open pit mines is a highly promising approach to overcome the limitations related to the transformation of the lower reservoir as a pit lake and the minimum required hydraulic head differences. This can be achieved due to the fact that the required topography has been established by mining-related excavations, while the economic re-utilization of former mine pits can substantially contribute to mine rehabilitation and environmental protection as well as to the public perception of HPHS projects.

The process of selecting appropriate sites for the construction and implementation of an HPHS system begins with the identification of possible constraints, parameters, criteria related to topography, the available and existing facilities, and environmentally vulnerable areas, as well as the technological and operational requirements that will render the system feasible and efficient. A review of the optimization and operation mode of hybrid power plants indicates that energy costs could be reduced by up to 47% [15].

Up to now, a few studies have focused on site selection for the construction of PHS systems by means of geographic information systems (GIS). For instance, the potential for transformation of typical hydropower systems, such as dams and other reservoirs, to PHS systems has been assessed at the country level using a GIS model [16], as well as at the regional scale [17] and for small-scale systems [18]. This has also been investigated in cases where there are already two existing reservoirs [19]. Other studies have combined GIS analysis with multi-criteria decision making (MCDM) to identify suitable sites for pumped hydro energy storage (PHES) systems, both at the country scale [20], as well as for large areas [21] and those exclusively located in natural environments exposing high elevation differences. However, there is a notable lack of studies referring to appropriate site selection for the construction of HPHS systems that integrate GIS analysis and the MCDM methodology within the areal boundaries of an abandoned open pit mine.

Moreover, the objective of this work was to assess the potential for energy storage in HPHS systems in former open pit mines based on the pre-defined location of the existing lower reservoir. Previous research [13,15,16] implemented at the country level has been strongly related to natural morphologic aspects. It is, therefore, evident that there is a need for more studies such as this one that focus on site selection for HPHS systems in open pit mines, especially related to abandoned coal mines by means of using integrated geospatial analyses. This study focuses on open pit mines where the rehabilitation or reclamation process has been planned to begin or is already ongoing, within the context of the current EU legislation that aims to transform Europe into the first climate-neutral continent in the world.

The applied methodology introduces a preliminary phase for research based on the existing landscape morphology related to the construction of the upper reservoir within a specific distance of the existing lower reservoir. In the scope of the present study, the objective of assessing HPHS potentials in open pit mines has been achieved for the first time by developing and applying a specific geospatial workflow that manipulates available data and specific geoprocessing algorithms. Here, the implemented criteria can be adjusted and integrated according to the specifications and recommendations for each case study while taking into account the required European and national legal and energy supply frameworks.

## 2. Materials and Methods

### 2.1. Study Area

The Kardia lignite mine is located in the Ptolemais Lignite Basin in Western Macedonia, Greece and is dominated by E–W trending normal Quaternary faults [22]. The Ptolemais basin covers a surface area of approximately 600 km^2^. The Ptolemais basin has a NW–SE direction, exceeding 20 km in length and width. The basin is filled with late Miocene to Pleistocene lake sediments, including intercalated lignite and alluvial deposits with a total thickness of up to 600 m [23].

The main lithologies of the sediments are sandy marls and clays, clayey marls, calcareous sands, and conglomerates, overlain by Quaternary conglomerates of terrestrial and fluvioterrestrial origin (Figure 3). The exposed stratigraphic sequence in the Kardia lignite mine belongs to the Early Pliocene Ptolemais formation. Based on the subsurface analysis [23], the lithology consists of lignite-marl alternations, intercalated with sands and silts, with an overall thickness of approximately 300 m [24,25,26]. The Ptolemais Basin is part of a tectonic trench over 250 km extending from northern Greece into North Macedonia [23,27].

According to previous literature, the thickness of the lignite-bearing layers (including intercalations) in the adjacent mining area ranges between 80–140 m at the western boundary of the mine near Mt. Askion. The thickness increases towards the SW, with 150 m of overlying lithologies. In the central and northwestern parts of the mine, the thickness of the overlying strata is about 20–60 m, and the thickness of the lignite seams varies [28].

Regarding the digital elevation model (DEM), the Kardia mine is an open and excavated area with surface elevations ranging between 460 m and 812 m (Figure 4) above sea level (a.s.l.) from the E to W direction, respectively (Figure 4).

Therefore, the DEM is frequently used to determine terrain attributes that include elevation at any point, slope, and aspect [29]. The DEM for the broader area of the mine was imported from satellite data using the Advanced Spaceborne Thermal Emission and Reflection Radiometer (ASTER), with a nominal horizontal accuracy of 15 and 20 m, respectively [23,30].

### 2.2. Literature Review on Site Criteria Selection

The existing literature on specific criteria for the selection of appropriate sites for the implementation of HPHS systems in former open pit mines is limited. There are a small number of studies that focus: (a) on the use of specific predefined criteria for site selection and (b) on geospatial models that can identify potentially suitable locations for the construction of new storage reservoirs. For example, the latter methodology has been used in an attempt to identify potential sites for the construction of new reservoirs in areas with already existing hydropower or water reservoirs [16]. In addition, similar approaches have been implemented for the assessment of the potential of pumped hydropower energy storage using two existing reservoirs in various European countries for proposed small-scale pumped hydro storage sites in mountainous areas [18]. In this context, two types of site selection criteria were identified: (a) criteria related to topographic aspects and (b) the proximity of the new reservoir to various components of the environment.

The first type of criteria considered the construction of the envisaged HPHS upper reservoir would be strongly related to the existing morphology, as well as to the distance between the proposed upper reservoir location and the existing lower one. The second category of selected criteria was based on proximity analyses on minimum distances from the investigated necessary entities, either natural or man-made, while taking into consideration the conservation of sensitive regions.

Regarding the topographic criteria, one of the most important for site selection is the average slope of the area where the upper reservoir is being investigated for its suitability. In the literature, an average slope angle of 0–5 degrees is used in existing GIS models [16,31], with an alternative of less than 5% [20]. The maximum slope angle of 5 degrees is established because the terrain needs to be as flat as possible for both technical and economic reasons [16]. In particular, the steeper the slope, the more excavation work will be required to level it, hence increasing the cost and the possibility of negative environmental consequences.

As for the minimum head (elevation difference) between reservoirs, there is a variety of arithmetical values in the bibliography, depending on the case study (Figure 5). For instance, many studies [16,19,20,31] have utilized a value of 150 m in PHS systems in Croatia, Turkey, and Iran, while [32] a 100 m elevation difference was suggested for another case in Turkey. Lu & Wang [17] have proposed a 500 m head, which was a recommendation by the Chinese government for a case in Tibet, while a 300 m head has been suggested for a proposed HPHS system to be installed on Skyros Island in Greece [33].

Regarding the minimum surface area of the upper reservoir, there is a restriction of 70,000 m^2^, and locations with smaller areas are not considered as appropriate for construction. According to [16,20,31], the proposed minimum area of 70,000 m^2^ includes 20,000 m^2^ for civil works, leaving a minimum area of 50,000 m^2^ for the upper reservoir.

The minimum depth of the upper reservoir (Figure 5), is most typically stated as 20 m [16,20,31]. In fact, the reservoir’s depth is proportional to its surface area and necessary volume; nevertheless, to reach a minimum volume of 1,000,000 m^3^ with a minimum surface area of 50,000 m^2^, a depth of at least 20 m is required. Although earlier studies used the 5 km threshold as the maximum distance between the upper and lower reservoirs [16,31], later studies have standardized this value to 20 km as a buffer zone, when searching for suitable sites in very large areas [19,20]. As for the minimum length of the water conduit that connects the two reservoirs, a value of 1500 m is mentioned in the literature [33]. According to [34], the length of the water conduit should be as short as possible to ensure a maximum elevation difference between the two reservoirs.

The second type of criteria, the proximity-based criteria, consist mostly of the minimum acceptable distances from various elements of the area surrounding the upper reservoir site. Taking into consideration the minimum distance from Natura 2000 regions, previous studies selected a 5 km radius along UNESCO sites [16,31]; however recent studies [19,20] set a radius zone of 500 m, where Natura 2000 conservation areas are totally excluded from consideration [19]. Concerning the minimum distance from populated/inhabited regions, the literature suggests a value of 500 m [16,19,20,31].

A minimum distance from natural bodies of water ensures avoiding any negative environmental impacts due to its operation, although the necessity of a water body (surficial or subsurface) in the vicinity of a HPHS site is evident.

The maximum distance from the existing power transmission grid has been defined as 20 km [19,31]. This value has been used as a constraint criterion, since a distance of more than 20 km would render the transformation of the existing infrastructure to a PHS system not viable [31], and it would also require the construction of new transmission lines [19].

The distance from existing tectonic lineaments is also an important criterion for site selection. The authors in [35] used the distance of potential upper reservoir sites from active geologic fault systems and fractured zones, as well as landslide areas, as a criterion for their assessment. There are no particular distance data provided, but each of the mapped features was graded based on its distance to the prospective construction locations. Lastly, the minimum distance from existing transportation networks (mainly roads and railways), was stated as either 100 m [31] or 200 m [16,19,20].

**Figure 5 sensors-23-00593-f005:**
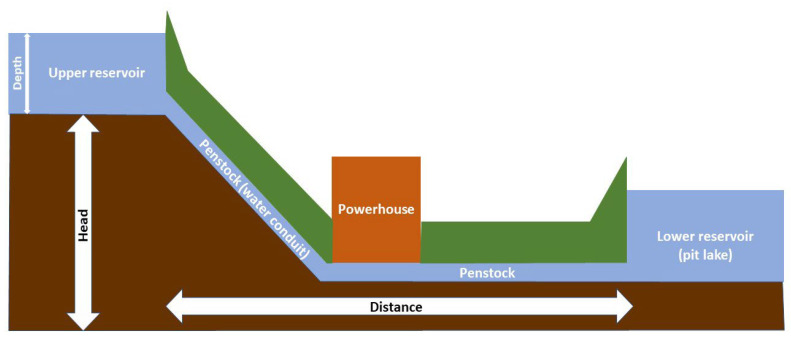
Schematic layout of a pumped storage plant (PSP). On the left, the hydraulic head elevation between the two reservoirs and the depth of the upper reservoir are illustrated. In addition, the distance between the two reservoirs and the length of the water conduit are depicted at the bottom and middle (modified from [36]).

The present study suggests a flexible GIS-based model investigating the transformation of an existing lower reservoir to an artificial lake by means of detecting suitable sites for the upper reservoir on a smooth area within the boundaries of an abandoned lignite open pit mine. The applied workflow can adapt different topology scenarios and integrate high-accuracy topographical and proximity-based data on the global scale to identify the HPHS potential.

The selected criteria highly affect the costs of HPHS system construction and operation, including environmental impact mitigation. The suggested upper reservoir locations are ranked according to their morphology, but detailed techno-economic studies are required for the final decision making towards the construction of the upper reservoir. In summary, the proposed GIS methodology can be applied in any open pit mine in the world by using the corresponding geospatial datasets.

### 2.3. Data

The datasets of this study are based on open-access products, such as data from the European Environment Agency (EEA) [37], the Open Street Map (OSM) [38], the Copernicus Land Monitoring Service (CLMS) [39], and unpublished data sources from the Public Power Corporation (PPC). The different varieties of the processed datasets, as well as their technical specifications, are outlined in Table 1.

### 2.4. Methodology

The workflow of this study is divided into a threefold process: (1) the development of a geodatabase; (2) the visualization and classification of the selected criteria; and (3) the ranking of the proposed areas within the boundaries of the study area. This process was derived in order to score the suggested areas for the construction of the upper reservoir adopting the analytic hierarchy process (AHP) approach.

According to the top part of the workflow (Figure 6), the pre-processing phase is the initial step where all geospatial data are obtained for import into the relational database. Digital elevation model (DEM) boundaries of the mine area and the location of the existing lower reservoir are needed for the first topographical analysis.

In the central part of the diagram, constraints are based on the acceptable flatness derived from the elevation datasets in order to calculate the total area that is suitable for the AHP analysis. In addition, with respect to datasets related to NATURA 2000, the maximum distance between reservoirs and the settlements category (Corine Land Cover 2018) were also used also as constraint criteria. Furthermore, after the implementation of the constraints, criteria were created in order be reclassified into a homogenized score from 1 to 5 within the boundaries of the identified sites. In the last part of the presented workflow (Figure 6), the final scoring of the proposed regions was applied using a multi-criteria decision making (MCDM) approach. Specifically, the AHP was carried out in the selected area of interest (AOI) that was defined by the Public Power Corporation of Greece (PPC). In particular, dataset specifications and technical information for the implemented methodology are described in detail in the following sections.

#### 2.4.1. Criteria and Constrains

In this work, the following geospatial datasets were used as inputs in order to construct the needed constraints and criteria according to Table 2. The criteria were also categorized according to their impact (positive or negative) based on the proximity analyses according to the final AHP ranking.

The constraints for Natura 2000, CLC 2018, and the drainage network were based on the generation of buffer zones to produce areas of 500 m search radii that were excluded from the geospatial analysis as unsuitable areas for the construction of the upper reservoir. Another constraint was the acceptable flatness, which was implemented initially to identify areas with appropriate slopes for the development of the upper reservoir. Lastly, the constraint of the minimum surface area was considered as a filter to determine the regions of at least 70,000 m^2^ size [16,20,31]. In accordance with the design criteria (Table 2), only 7 of the 98 polygons generated were generally considered acceptable for the assessment of potential locations for the AHP ranking. Moreover, the creation of criteria was based on the proximity of each spatial feature to potentially suitable sites, which was calculated using the following Euclidean distance Equation [40]:(1)$$d\left(x,x^{\prime }\right)=\sqrt{{\left(x_{1}-z_{1}\right)}^{2}+{\left(x_{2}-z_{1}\right)}^{2}+\ldots +{\left(x_{n}-z_{n}\right)}^{2}}=\sqrt{\overset{n}{\underset{i}{\sum }}{\left(x_{i}-z_{i}\right)}^{2},}\;$$
where *x_i_* is the coordinate for the *x* location and *z_i_* is for the z location.

In addition, the average head (h) difference was calculated as a criterion, due to the difference between the average elevation of each site and the existing lower reservoir. Specifically, a more detailed description of the selected factors is presented in the following paragraphs.

1.The existing location of the lower reservoir

The distance between the upper and the pre-determined location of the existing lower reservoir (Figure 7) is a critical aspect in terms of frictional losses in the water conduits and, thus, operational costs. In this study, the location of the lower reservoir was considered as the area with the lowest elevation within the boundaries of the open pit. A buffer zone of 5 km was used as the threshold for the maximum distance between the two reservoirs, whereby shorter distances had a more positive impact in the analysis and were ranked as more suitable for the upper reservoir construction.

2.Topography

Topographic parameters were derived from a high-resolution DEM created with contour lines and elevation points provided by the PPC. Initially, the produced DEM was corrected using the Fill algorithm to remove possible sinks and peaks. The Fill tool uses the equivalents of several tools, such as focal flow, flow direction, sink, watershed, and zonal fill, to locate and fill the sinks [41,42]. Open data sources of the Shuttle Radar Topography Mission (SRTM) [43], and the DEM of Europe (EU-DEM) [44] were also investigated; however, their technical specifications didn’t meet the requirements of this study, due to their unknown sensing periods and low spatial resolutions.

Furthermore, the slope angle of the area under examination was considered as a constraint to identify locations that were suitable for the construction of the upper reservoir. The maximum slope angle threshold of 5 degrees was selected as acceptable flatness [16] of the site’s morphology. Specifically, areas with higher slope angles were excluded from the geospatial analysis. In addition, the elevation difference was used as a criterion between the average head elevation of the defined lower reservoir and proposed suitable regions (Figure 7). In order to calculate the average elevations, the zonal statistics tool was used in the GIS environment within the study area’s boundaries. Particularly, the delineated zones represent the polygons of accepted flatness, where the calculation of the average elevation was based on the corrected DEM of the AOI.

3.Corine Land Cover

In 1985, the CORINE Land Cover (CLC) inventory was established (reference year 1990). There were revisions in 2000, 2006, 2012, and 2018. It consists of a list of 44 classes of land cover. The CLC employs a minimum mapping unit (MMU) of 25 hectares (ha) for areal phenomena and 100 m for linear phenomena. Regarding the CLC dataset of 2018, the man-made category of settlements was filtered out for the broader area. The nearest settlement to the study area is Pontokomi, whereby according to the PPC, this settlement has been expropriated, but the region will not be used in future for surface mining activities. The next nearest settlement is Mavrodendri, which is located at a distance of more than 4 km.

4.Natura 2000

According to the European Environmental Agency (EEA), Natura 2000 is a network of sites designed under the Birds Directive and the Habitats Directive, which contain regions protected in their own right, such as breeding and resting areas for rare and threatened species, and natural habitat types, both on land and in the marine environment. The Natura 2000 database is composed of the 27 EU member countries and submitted by their national authorities in a specific data format containing an extensive description and borders (spatial data) of the respective sites and their ecologies [45]. The nearest Natura 2000 sensitive region is more than 10 km away from the present study area.

5.Transportation network

The construction of the upper reservoir must be located as close to the existing transportation network due to the lower cost. Under this light, the proximity to a transportation network has a positive impact in the geospatial analysis. In this study, only the main network (Figure 8) was utilized for the criterion generation. Particularly, the closer the distance to roads, the higher the area’s suitability score. The data were collected from the OSM and used for the assessment of the criterion “Distance to transportation network”.

6.Tectonic lineaments

Another important criterion is the distance to tectonic elements, such as geologic faults (Figure 8) and the spatial distribution of geologic fracture networks. The proximity of the upper reservoir to these elements has a negative impact on the analysis, due to the potential exposure of the HPHS system to geotechnical ground instability. Data on existing geological faults were taken from [23] and used for the creation of the criterion “Distance to the geological faults “. It should be mentioned that the only available information was the linear fault traces, with no additional information on their type, activity status, and geometry.

7.Power transmission grid

The power transmission grid (Figure 8) is crucial to relay the produced energy from the HPHS system to the grid and to supply the required energy for the electrical equipment of the system. Available data were provided by the PPC, modified, and validated using the Google Earth Pro platform. Suitability affected positively in proportion to the degree to which electric grids were adjacent to the location of the HPHS power house. According to this, the estimation of the criterion “Distance to power transmission grid” was implemented.

8.Drainage network

In order to mitigate the potential negative impact on the natural environment, the upper reservoir should be located as far as possible from the existing drainage network (Figure 8). Buffer zones of 500 m were assumed around rivers to exclude them from the analysis. The closer the distance to rivers, the more negative affects the respective area’s suitability score was. The data were collected from an open-source geoportal [45] and applied for the assessment of the criterion “Distance to the drainage network”.

To make a ranking system for the criteria, all datasets were reclassified into a single tactical scale to become comparable to each other in the AHP analysis. Due to the limitations in the literature regarding the classification of each criterion, the natural break (Jenks) method [46] was implemented, which is characterized as the most appropriate for the classification of values into classes [47].

The resulting classes illustrated the suitability of every location regarding the cost distance and the environmental and hazard assessment. Particularly, five classes (Table 3) correspond to the degree of association of each variable with the suitability of the proposed areas for the implementation of the upper reservoir, where 5 represents the highest suitability, 4 represents high suitability, 3 represents moderate suitability, 2 represents low suitability, and 1 represents very low suitability.

#### 2.4.2. Analytical Hierarchy Process (AHP)

The AHP method, developed by Thomas Saaty in 1978 [48] is a decision-making tool that deals with multi-criteria evaluation. AHP has been utilized in various scientific fields, due to its flexibility [49], and can also be applied to integrated geospatial analyses.

The first step on the AHP is the determination of the objective, criteria and alternatives, which is the main part of the decision-making process since it structures the decision problem as a hierarchical structure diagram (Figure 9).

This study aims to classify the proposed sites into a ranking system according to the aforementioned criteria. Particularly, the most important criteria for the alternatives were the average head elevation difference and distance between reservoirs based on the literature and the expert’s judgment.

Precisely, the AHP is a pairwise comparison approach that decomposes problems into hierarchical systems to support decision making. It is based on complex calculations using matrix algebra that provide a numerical scale that ranges from 1 to 9 to calibrate the quantitative and qualitative performances of priorities [50]. The fundamental scale of comparison proposed by Saaty [51] was used to compare pairings (Table 4). As a result, the final pairwise comparison creates a 6 × 6 table, in which the diagonal comparisons that represent the pairwise result of the factor itself are equal to 1. The criteria were selected based on the literature, while comparative values were assigned in accordance to the judgement of industrial experts of the PPC with expertise in coal mines and specifically in the field of mining and geoengineering, geology and hydrogeology, industrial safety, environmental engineering, and sustainable energy technologies. As a result, a hierarchical evaluation model was derived, with an AHP pairwise comparison scale, as described hereinafter regarding the relevance of each criterion in relation to all other criteria.

The final weights in the output process were calculated due to their influence on the study problem [52]. They were computed by the following technique, which utilizes the geometric mean of each line (*u_i_*) and divides it by the sum of the geometric mean of all rows (*u_k_*) of the matrix, thus calculating the weights of importance of each factor as given in Equation (2) [52]:(2)$$w_{i}=u_{i}/\overset{n}{\underset{k=1}{\sum }}u_{k},\;$$

Thus, these are the values of the importance of Saaty’s fundamental scale that are presented in the following Table 4.

The consistency of the pairwise comparisons of the square table is evaluated by the consistency ratio (*CR*), which is obliged to be less than 0.1 and greater than 0. Specifically, in this study, the *CR* was calculated with an acceptable consistency at 0.07. The *CR* value is calculated by Equation (3):(3)$$CR=\frac{CI}{RI},\;$$
where the *RI* is the randomness index value that depends on the order of the matrix published by Saaty [50], and *CI* is the consistency index, which is calculated from the following Equation (4):(4)$$CI=\frac{\left(\lambda_{max\;}-n\right)}{\left(n-1\right)}\;,\;$$
where *λ_max_* is the largest eigenvalue of the matrix and *n* is the order of the matrix.

The calculated weights were used as multiplier factors on each classified criterion. Each weight value corresponds to the rank of importance of each criterion. Finally, a map is generated by calculating the cumulative of all multiplied criteria used in the present study.

## 3. Results

### 3.1. Criteria Ranking

In this study, six criteria were selected to be homogenized and used as inputs in the presented AHP implementation (Figure 10).

According to Figure 10, the suitability ranking of each potential area was visualized with different colors in terms of suitability. Specifically, with dark green color were regions with very high suitability, followed by green colored areas with high suitability, yellow colored areas with moderate suitability, orange colored areas with low suitability sites, and red colored areas with very low suitability for the construction of the upper reservoir.

A multi-criteria decision making (MCDM) technique was applied in order to determine the suitability score for the suggested regions for the construction of the upper reservoir. As mentioned before, the AHP methodology was used to assess and evaluate scores based on the selected criteria related to the suggested sites. The pairwise comparison of the implemented criteria is illustrated in Table 5, which entailed a hierarchical evaluation of the relevance of each criterion in relation to all other criteria. Particularly in this work, the AHP was used to rank the suggested sites from best to worst by utilizing the following weights.

According to the calculation of the weights conducted by the criteria comparison, the most important factor for the construction of the upper reservoir was the average elevation difference (0.37). The next important factor was the distance between reservoirs (0.25), followed by the distance to faults (0.19), and the distance to rivers (0.09), while the lowest weights of important were related to the distances to the power transmission grid (0.04) and the distance to the transportation network (0.03). Each weight value corresponded to the rank of importance of each criterion and was utilized as a multiplier factor on each ranked criterion, respectively.

The MCDM technique was applied within the boundaries of suitable areas using AHP by overlaying the results of the ranking criteria to classify the study area into a scale that ranged between 0 and 4.69, with the highest values representing the most suitable areas for the upper reservoir construction. The results of the AHP and GIS analyses (Figure 11) showed that the regions with higher scores (dark green colored) were located in the center of the study area at the labeled sites “1” and “3”, while the lower scores (red color) were detected at site “0” at the north side of the AOI.

These highly scored areas were characterized by maximum average elevation differences and relatively close proximities to the lower reservoir. Particularly, the results within the boundaries of seven regions indicated that 9.27% (212,884 m^2^) of the regions had very low suitability for the upper reservoir construction, 15.83% (363,599 m^2^) had low suitability, 23.99% (550,998 m^2^) had moderate suitability, 24.99% (573,813 m^2^) had high suitability and 25.92% (595,125 m^2^) had very high suitability.

### 3.2. Storage Energy Capacity Estimation

According to [16,18,29] the energy storage capacity for a prospective HPHS site can be calculated from Equation (5):(5)E=p×g×h×V×η, 
where, *p* is the density of water (1019 kg/m^3^), *g* is the acceleration of gravity (9.81 m/s^2^), *h* is the head, *V* is the volume of water in the upper reservoir, and *η* is the efficiency of the pump/turbine unit (assumed as 90%). The volume of the upper reservoir was estimated by *V* = (*A* − 20,000) * *d*, where *A* is the area of the discovered location and *d* is the depth of the new reservoir, which was assumed to be 20 m. According to the literature, 20,000 m^2^ is the area that is considered for civil work and should be subtracted from the total area. The estimation of the storage capacity of every site is presented and analyzed in the following subsection (Table 6).

### 3.3. Statistical Analysis of the Proposed Areas

The quantification of the suitability results related to the GIS analysis and the storage capacity estimation are displayed in the following Table 6.

The average AHP score was calculated for each location using the zonal statistics tool and is actually the average value of the AHP analysis for each location.

The distance between the two reservoirs is the Euclidean distance between the centroid of the lower reservoir and the centroids of any suggested location. Particularly, Figure 12a reveals that there was a strong correlation between the average AHP score and average head difference, while Figure 12c shows that the average AHP score was not strongly dependent on the distance of each lower reservoir location.

## 4. Discussion

Geospatial analysis is a promising tool that can be used by policymakers and stakeholders for decision making with regard to the implementation of HPHS systems in abandoned open pit mines, for example, in the context of spatial development, or the optimum areal utilization for future constructions that can mitigate the financial costs, environmental impacts, and exposure to hazards, such as landslides, earthquakes, and floods. Additionally, it is a useful tool to maximize energy storage by calculating the best-fit options to meet criteria selected according to the specific demands of the end user. The spatial results of the multi-criteria decision making (MCDM) analysis can provide the possibility to analyze different scenarios and designate suitable areas for the development of space in open pit mines. This work generated a semi-automatic workflow to determine the most suitable areas for the construction of an upper reservoir for a HPHS implementation in the Kardia open pit mine located in the Ptolemais Basin (Greece).

The closure of coal production and related abandonment of coal mines in Europe raises the issue of the management of the former mining regions and regional economy most affected by the ceased mining operations. Planned decommissioning of lignite mining requires innovative and economical strategies to support European coal regions in transition. The repurposing of open pit mines into HPHS for excess energy storage from the electric grid and renewable sources will contribute to the EU Green Deal, increasing the economic value, supporting the regional job market, and securing the EU energy supply.

Moreover, this study presented an AHP approach integrated through GIS and applied for a potential HPHS facility in the already closed open pit lignite mine of Kardia, North Greece. It combined the most important criteria derived from the literature in addition to industrial experts’ judgement, with expertise in coal mines and specifically in the field of mining and geoengineering, geology and hydrogeology, industrial safety, environmental engineering, and sustainable energy technologies.

Due to the dynamic evolution of the mine excavation environment, the study also highlighted the limitations of existing open-source datasets. The dynamic evolution of the excavation environment tends to make the applicability of already existing open-source DEM unsuitable for the implementation of an analysis that corresponds to the recent morphology of open pit mines. Under this aspect, this work suggests the acquisition of an updated DEM from the mine operators or their generation by using the interferometric synthetic aperture radar (InSAR) technique. Furthermore, the classification of the criteria into a single tactical scale using the natural break method is a holistic approach that can be useful when the literature data is limited; however, the use of specific numerical ranges on each criterion will ensure more accurate and suitable results.

In addition, the undertaken statistical analysis shows that the storage capacity was not a factor of the average AHP score, which was mainly dependent on the coverage area of each site, which highlights that the determination of the location for the construction of the upper reservoir is a complex process that requires the combination of the GIS results with additional data. In particular, HPHS systems necessitate additional in-depth research in the context of hydrogeologic, hydrogeochemical, and geotechnical concerns that may arise as a result of variations in the water level in both reservoirs as well as its chemical composition in the presence of pyrite oxidation. Despite these caveats, HPHS is expected to exhibit a substantially lower environmental footprint, together with lower economic costs, in comparison with the construction of new reservoirs.

In summary, the outcome of the present study shows that suitable locations within the boundaries of the open pit coal mine covered about 36% of the total study area, whereby the top-ranked regions were located at the highest altitude areas covering up to 9.47% of the study area. Furthermore, the Kardia lignite mine lies within a favorable distance from the electricity transmission network, which is a crucial advantage.

The results for the seven selected sites indicated that 9.27% (212,884 m^2^) of the investigated area had very low suitability, 15.83% (363,599 m^2^) had low suitability, 23.99% (550,998 m^2^) had moderate suitability, 24.99% (573,813 m^2^) had high suitability, and 25.92% (595,125 m^2^) had very high suitability for the upper reservoir construction. It should be highlighted that, concerning the most suitable locations for the implementation of the HPHS upper reservoir, those derived from this study are in accordance with the results provided from the unpublished feasibility studies of the HPHS in the Kardia mine. More specifically, two of the proposed sites regarding the applied methodology have been recently suggested in the pre-feasibility studies, which are currently in progress.

## 5. Conclusions

In this study, an innovative, semi-automatic workflow was introduced to identify potentially suitable areas for the construction of an upper reservoir for an HPHS system to be implemented in the Kardia open pit coal mine (Western Macedonia, Greece) that can also be applied to other open pit mines worldwide. Additionally, an initial estimation of the energy storage capacity was given for the prospective HPHS site. This study was conducted by using a GIS model and the support of industrial experts’ judgement from the PPC, with expertise in coal mines and specifically in the field of mining and geoengineering, geology and hydrogeology, industrial safety, environmental engineering, and sustainable energy technologies, to classify six geospatial criteria that were identified as the main factors for the selection of suitable reservoir locations. These criteria were related to the cost distance as well as environmental and natural hazard assessment. According to the MCDM results of the criteria classification and their quantification, the average hydraulic head difference, which was defined as the most important factor, had a decisive role in the determination of the location of the upper reservoir. As such, the two areas with the highest scores were located at the highest elevation values, while the site with the lowest score had the furthest distance from the lower reservoir and was adjacent to the geological faults. Taking into account the available literature data and the results of the present case study, we conclude that the average elevation difference, combined with specific requirements and the demands on storage capacity, play an important role in the selection of the most suitable sites for the construction of the upper reservoir. Future work should focus on the utilization of geotechnical criteria related to slope stability and other earth observation products, such as the monitoring of surface deformation (InSAR), not only during the pre-construction phase, but also while the HPHS system is in operation mode.

Greece is highly integrated with solar and wind energy sources that require energy storage. The application of HPHS systems in abandoned open pit mines as storage reservoirs utilizing existing lower water reservoirs or pit lakes, into which water can be released when additional electricity is required, is a highly promising means of balancing the energy demand annually and mitigating the rising cost of batteries. Excess renewable energy can be used to pump water into the upper reservoir when energy demands and market costs are low, creating a relatively closed system with small energy loss. Moreover, the transformation regarding the upper reservoir construction within the boundaries of the abandoned coal mine can prevent further environmental impacts and contribute to a smooth restoration of land. The implementation of the HPHS systems is a complex issue in open pit mines with many challenges related to HPHS design, operation, and monitoring in open pit mines. The suggested workflow can contribute to the successful and comprehensive management of open pit mines at the stage of pre-feasibility studies and analyses.

In the context of sustainable development towards the green energy transition, this work introduces an innovative tool that can identify the initial spatial planning globally for an HPHS system based on the latest morphological landscape within the boundaries of open pit mines.

## Figures and Tables

**Figure 1 sensors-23-00593-f001:**
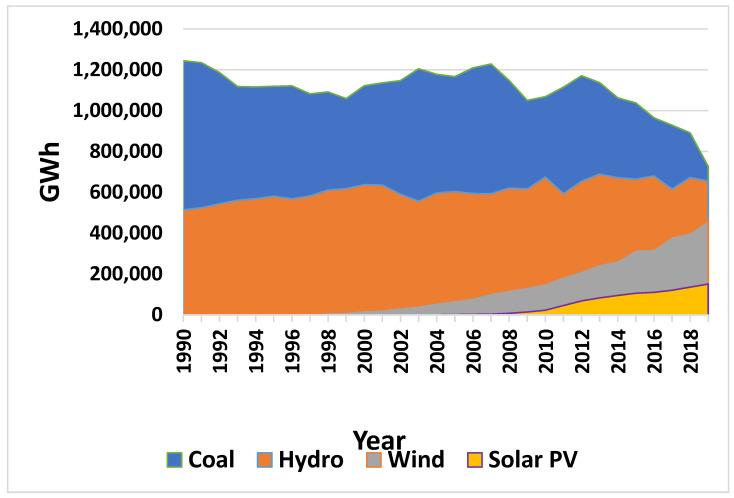
Europe’s electricity generation from 1990 to 2021 (modified from International Energy Agency ΙΕA [3]).

**Figure 2 sensors-23-00593-f002:**
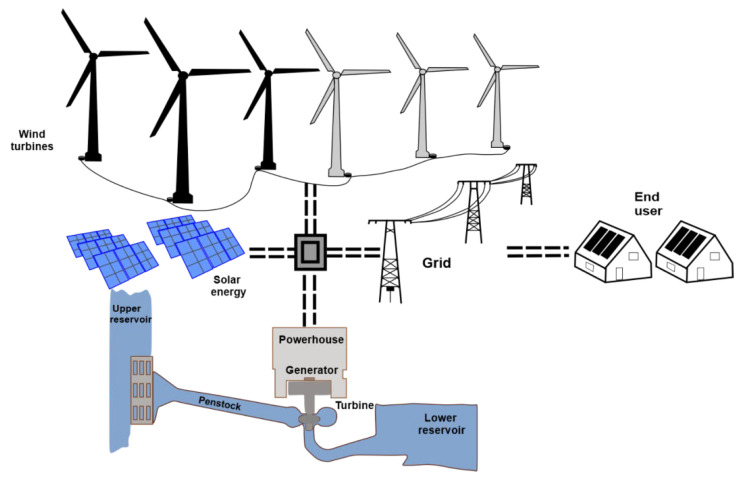
Schematic of hybrid pumped hydro storage system powered by excess energy from renewable energy sources and the electric grid.

**Figure 3 sensors-23-00593-f003:**
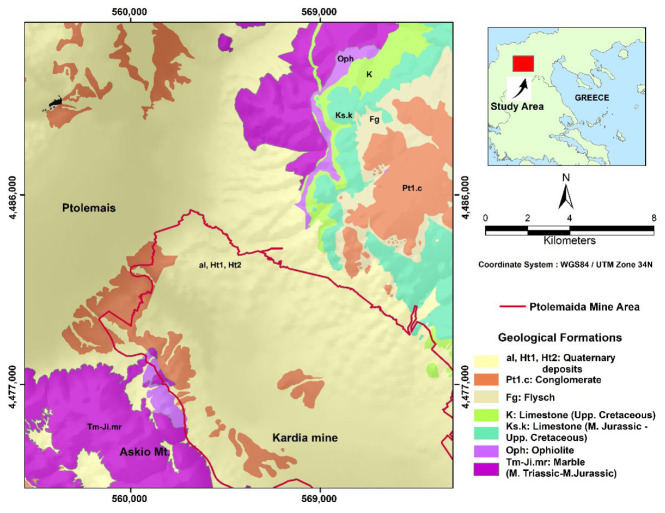
Simplified geological map of the Ptolemais area, located SE of the city of Ptolemais (based and modified on IGME geological sheet, Ptolemais 1:50,000 [23,24]).

**Figure 4 sensors-23-00593-f004:**
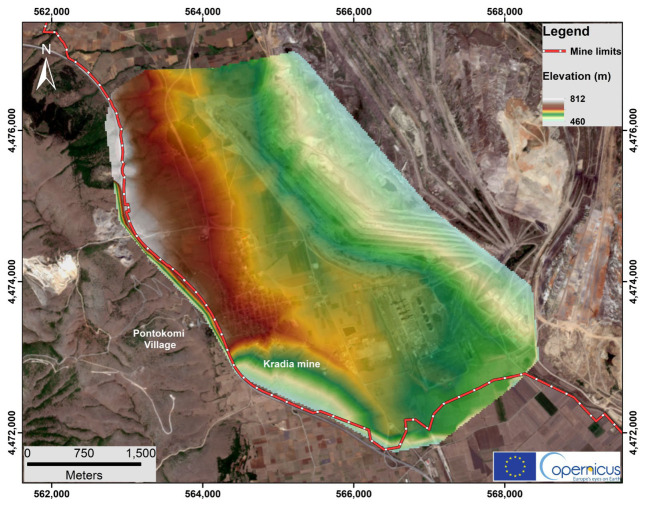
Digital elevation model in the broader study area of Kardia mine using ArcGIS 10.4 version. The base map is provided by the Sentinel 2b satellite European Union Copernicus program with sensing date 31 August 2021 [23].

**Figure 6 sensors-23-00593-f006:**
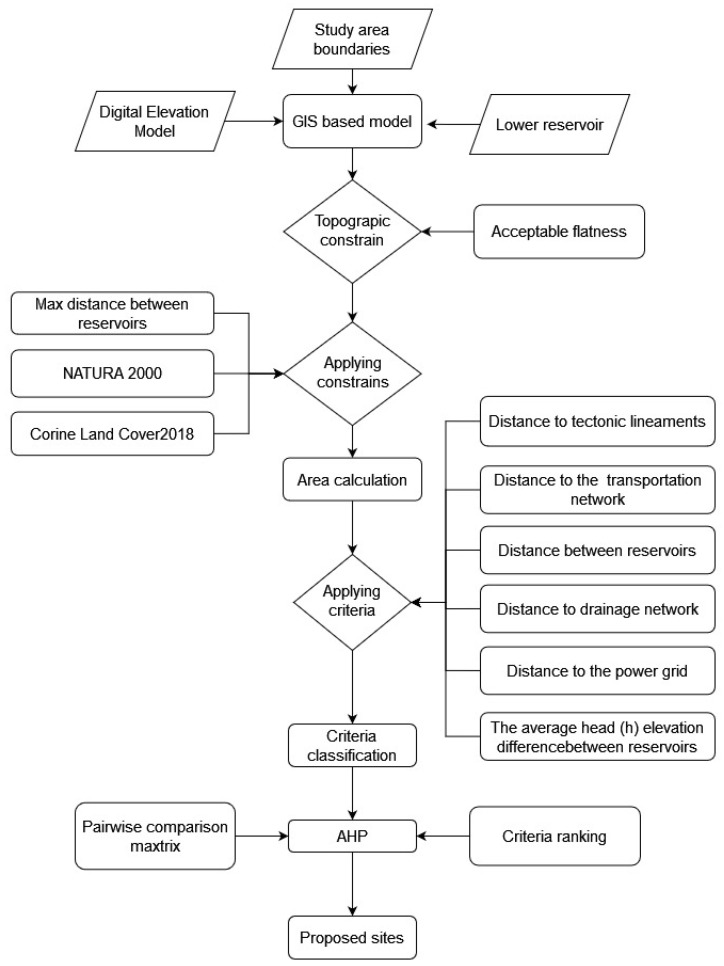
Schematic workflow of the geoprocessing tools that were implemented utilizing various datasets and adopting the analytical hierarchy process (AHP) approach.

**Figure 7 sensors-23-00593-f007:**
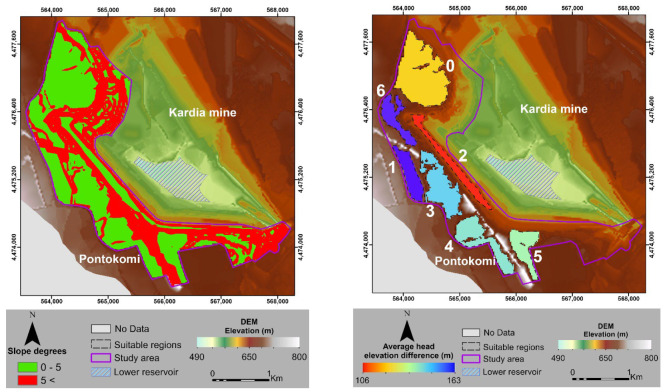
Constraint of slope analysis (**left**), where green color corresponds to the areas of acceptable flatness and red color to unsuitable ones. Average head elevation difference (**right**), where blue color illustrates the highest differences and red color the lowest ones.

**Figure 8 sensors-23-00593-f008:**
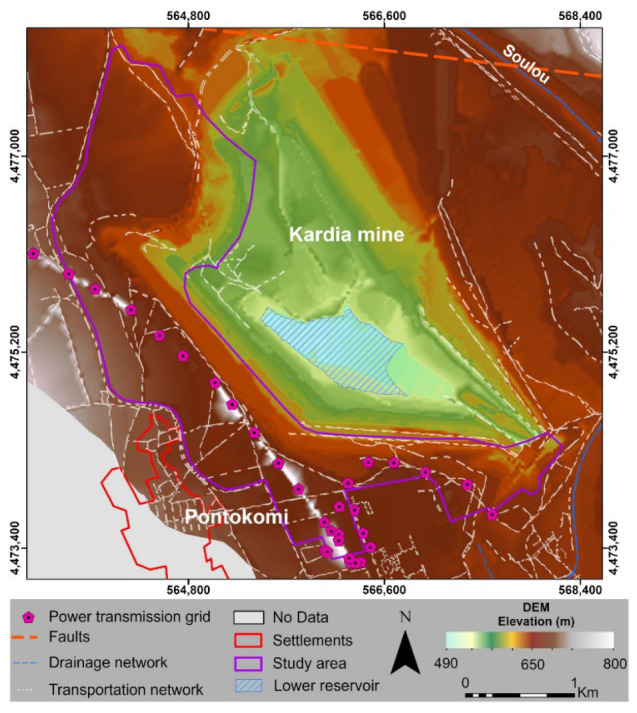
Map of Kardia mine illustrating the data inputs of geospatial analysis, e.g., the orange dashed lines depict the geologic faults, white lines present the transportation network, purple polygons show the power transmission grid, blue lines indicate the drainage network, the red polygons illustrate existing settlements, and the blue-dashed polygon area depicts the lower reservoir location.

**Figure 9 sensors-23-00593-f009:**
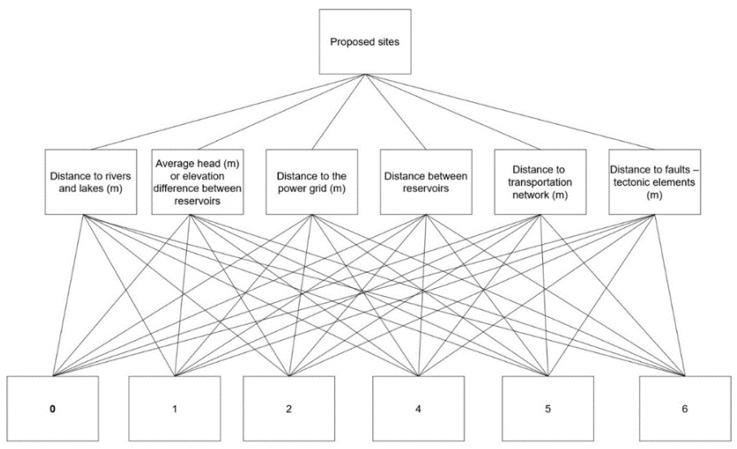
Hierarchy structure for upper reservoir site selection.

**Figure 10 sensors-23-00593-f010:**
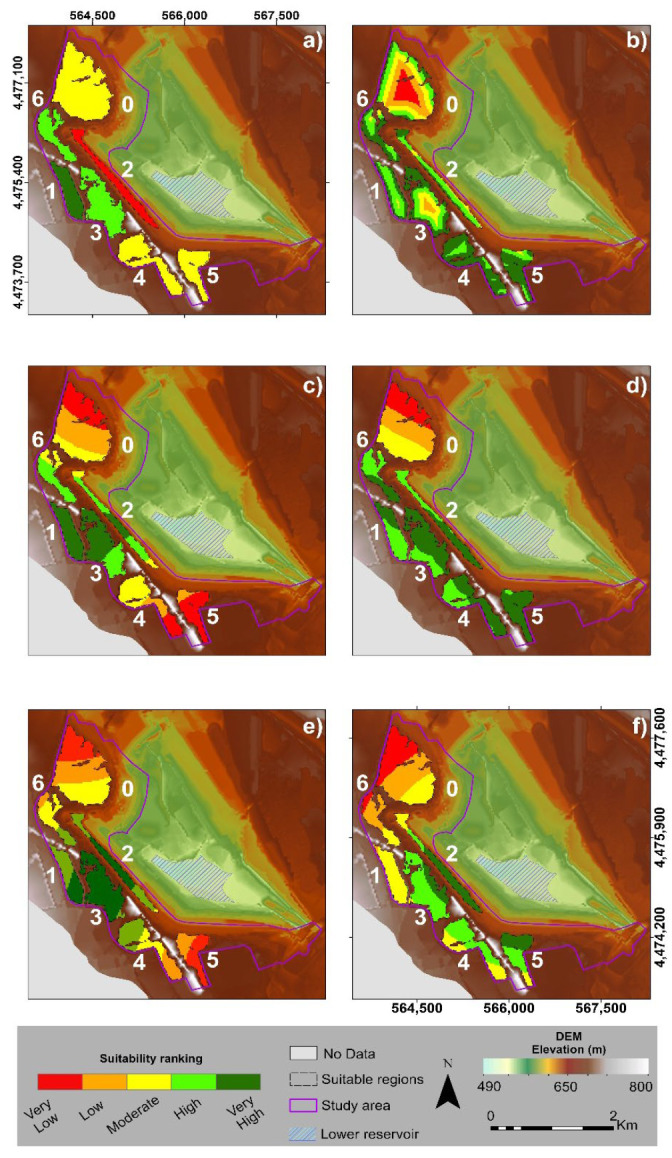
Criteria ranking maps illustrating the suitability score from very low (red color) to very high (dark green colors) regarding the HPHS proposed sites according to the (**a**) average head elevation difference, (**b**) distance from transportation network, (**c**) distance from geological faults, (**d**) distance from power transmission grid, (**e**) distance from drainage network and (**f**) distance from lower reservoir.

**Figure 11 sensors-23-00593-f011:**
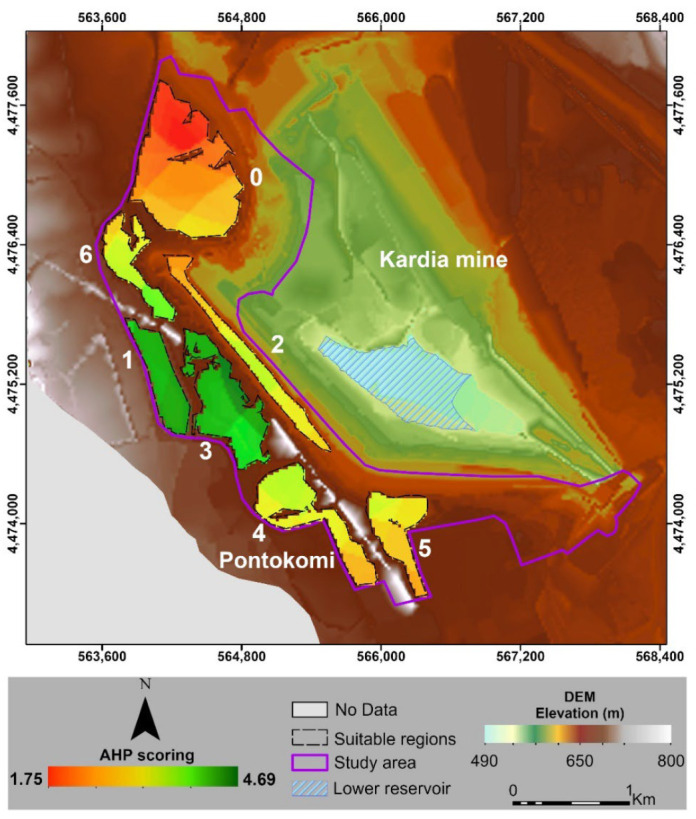
Ranking suitability map for the construction of upper reservoir using AHP method. The green color illustrates the areas with the higher suitability score and red color the lower, respectively.

**Figure 12 sensors-23-00593-f012:**
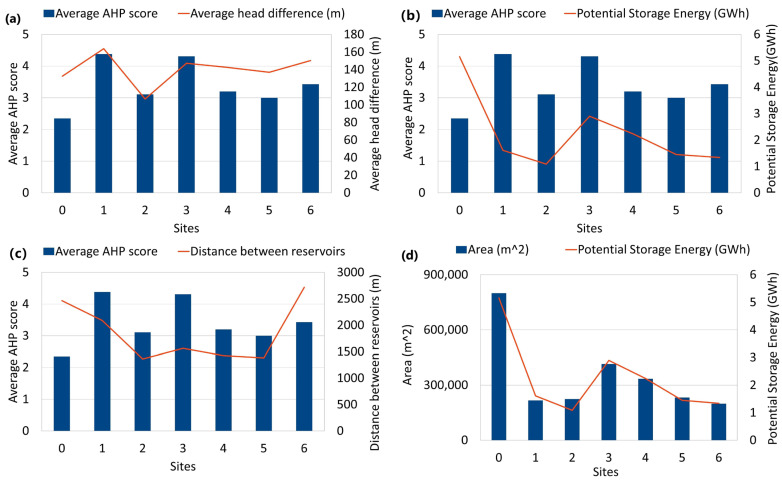
Correlation diagrams between (**a**) the average AHP score versus average head difference (m), (**b**) average AHP score versus potential storage energy (GWh), (**c**) average AHP score in comparison with distance between the two reservoirs, (**d**) potential storage energy (GWh) versus areal coverage.

**Table 1 sensors-23-00593-t001:** Descriptive characteristics of the processed datasets.

Dataset	Type	Source	Scale
**Contour lines**	Vector file/polyline	PPC	1:25,000
**Elevation points**	Vector file/points		1:25,000
**Land Cover/Land Use**	Vector file/polygon	CORINE Land Cover 2018	
**Natura 2000**	Vector file/polygon	EEA	
**Transportation Network**	Vector file/polygon	Open Street Map	
**Geological Faults**	Vector file/polyline	Literature	1:50,000
**Drainage network**	Vector file/polyline	EEA	1:100,000

**Table 2 sensors-23-00593-t002:** Design criteria and constrains.

Factor	Type	Impact	Criterion Attribute
**Location of the existing lower reservoir**	Criterion/Constraint	Positive	Distance between reservoirs
**Topography**	Criterion/Constraint	Positive	The average head (h) elevation difference between reservoirs & Acceptable Flatness (degrees)
**Natura 2000**	Constraint	-	Minimum distance to nature conservation, landscape protection areas, and natural habitats; minimum distance to populated areas
**Land c/land use** **Corine 2018** **(Settlements subcategory)**	Constraint	-	
**Transportation network**	Criterion	Positive	Distance to the transportation network
**Tectonic lineaments**	Criterion	Negative	Distance to lineaments
**Power grid**	Criterion	Positive	Distance to the power grid
**Drainage network**	Criterion/Constraint	Negative	Distance to drainage network

**Table 3 sensors-23-00593-t003:** Criteria classification ranking values.

Ranking Classes	1	2	3	4	5
**Average head (h) or elevation difference between reservoirs (m)**	106	106–131	131–141	141–149	149–163
**Distance between reservoirs (m)**	2019–2621	1613–2019	1223–1613	867–1223	460–867
**Distance to the power grid (m)**	1432–1835	1115–1432	690–1115	309–690	0–309
**Distance to transportation network (m)**	325–475	230–325	150–230	78.23–150	0–78
**Distance to tectonic lineaments (m)**	371–1140	1140–1557	1557–2059	2059–2572	2572–3096
**Distance to rivers and lakes (m)**	1075–1951	1951–2727	2727–3252	3252–3703	3703.35–4266

**Table 4 sensors-23-00593-t004:** Saaty’s scale of importance intensities [51].

Intensity of Importance on an Absolute Scale	Definition	Explanation
**1**	Equal importance	Two factors contribute equally to the objective.
**3**	Moderate importance of one over another	Experience and judgment slightly favor one over the other.
**5**	Essential or strong importance	Experience and judgment strongly favor one over the other.
**7**	Very strong importance	Experience and judgment very strongly favor one over the other. Its importance is demonstrated in practice.
**9**	Extreme importance	The evidence favoring one over the other is of the highest possible validity.
**2, 4, 6, 8**	Intermediate values	When compromise is needed.

**Table 5 sensors-23-00593-t005:** Weights of the selected criteria for the AHP analysis.

	Average Head (m)	Distance between Reservoirs (m)	Distance to the Power Grid (m)	Distance to Existing Transportation Network (m)	Distance to Faults (m)	Distance to Rivers and Lakes (m)	Sum (U_i_)	Weights
**Average head (m)**	1.00	2.00	7.00	7.00	3.00	5.00	2.2681	0.3780
**Distance between reservoirs (m)**	0.50	1.00	6.00	7.00	2.00	4.00	1.5476	0.2579
**Distance to the power grid (m)**	0.14	0.17	1.00	2.00	0.20	0.25	0.2686	0.0448
**Distance to transportation network (m)**	0.14	0.14	0.50	1.00	0.20	0.25	0.2030	0.0338
**Distance to faults (m)**	0.33	0.50	5.00	5.00	1.00	5.00	1.1461	0.1910
**Distance to rivers and lakes (m)**	0.20	0.25	4.00	4.00	0.20	1.00	0.5667	0.0945

**Table 6 sensors-23-00593-t006:** Statistical results of AHP scoring for the suggested upper reservoir locations.

Site	Average AHP Score	Area (m^2^)	Volume (m^3^)	Average Head Difference (m)	Potential Storage Energy (GWh)	Distance between Reservoir (m)
**0**	2.35	799,513	15,590,260	132.48	5.16	2465
**1**	4.38	217,171	3,943,420	163.82	1.61	2089
**2**	3.11	224,727	4,094,540	106.71	1.09	1363
**3**	4.31	415,196	7,903,920	147.19	2.90	1563
**4**	3.20	334,787	6,295,740	142.48	2.24	1425
**5**	3.00	232,247	4,244,940	137	1.45	1381
**6**	3.43	198,592	3,571,840	150.30	1.34	2717

## Data Availability

Data available upon request.

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
