# Peer review of "GIS-Based Assessment of Hybrid Pumped Hydro Storage as a Potential Solution for the Clean Energy Transition: The Case of the Kardia Lignite Mine, Western Greece"

_sensors, 2023, doi:10.3390/s23020593_

Round 1

Reviewer 1 Report

Dear Authors,

thank you for your great work.
Below you can find some minor comments which can be useful to improve the quality of your paper:
- The quality of all the figures is poor. In particular, figure 6 seems to be very impactful for the overall methodology and the scheme could be improved.
- Authors might consider the integration of real-time data coming from the  global monitoring for environment and security geospatial database Copernicus Programme
- It would be useful to specify how many engineering and geology experts have been involved in the workshop. However, experts coming from different organization would have been providing more consistency to the research.

Author Response

Thank you for the opportunity to submit the revised version of the paper " GIS-based assessment of Hybrid Pumped Hydro Storage as a potential solution for the clean energy transition: The Case of Kardia Lignite Mine, Western Greece" for consideration by the Sensors Journal. We are grateful about the time and effort that you dedicated to providing feedback on our manuscript. We are also glad about the helpful comments and the suggestions made to improve our manuscript. Please see the following section for a detailed response to the reviewers' comments and concerns.

Reviewer 1

  1. The quality of all the figures is poor. In particular, figure 6 seems to be very impactful for the overall methodology and the scheme could be improved.

Author’s response: Dear reviewer, thank you for your comment. We revised the quality of all figures accordingly in order to be improved.

  1. Authors might consider the integration of real-time data coming from the global monitoring for environment and security geospatial database Copernicus Programme.

Author’s response: Dear reviewer, thank you for your suggestion. However, the main limitation is related to the recent Digital Elevation Model (DEM), due to the fact that the methodology is based on the latest topographical features. In the future work we plan to integrate and validate more recent datasets from Copernicus database which is continuously growing. Also, the main part of the data processing regarding the GIS methodology was implemented during the first period of the project as a preliminary phase for the Hybrid Pumped Hydro Power Storage (HPHS) suitability analysis.

  1. It would be useful to specify how many engineering and geology experts have been involved in the workshop. However, experts coming from different organization would have been providing more consistency to the research.

Author’s response: Dear reviewer, thank you for the pointing this out. The selected methodology and specific criteria were based on the judgement of industrial experts of PPC, with expertise in coal mines and specifically in the field of mining and geoengineering, geology and hydrogeology, industrial safety, environmental engineering and sustainable energy technologies. Furthermore, the whole analysis was carried out in the context of ATLANTIS (RFCS) European research program.

Reviewer 2 Report

1.      Figure 1, Figure 3, Figure 4, Figure 5 , Figure 6 and so on : In all these figure, the text is blurred and not readable. Change all the pictures with higher resolution.

2.      Introduction is little weak. Please improve and clearly specify the research gap and novelty of the paper.

3.      Check and improve the technical English. Break the long sentences.

4.      Conclusion: Provide crisp concluding points of the work.

5.      How the results are validated?

Author Response

We would like to thank you for the opportunity to submit the revised version of the paper " GIS-based assessment of Hybrid Pumped Hydro Storage as a potential solution for the clean energy transition: The Case of Kardia Lignite Mine, Western Greece" for consideration by the Sensors Journal. We appreciate the time and the resources that you have spented in providing us feedback on our paper. For a point-by-point response to the reviewers' comments and concerns, please see the following section.

Reviewer 2

  1. Figure 1, Figure 3, Figure 4, Figure 5, Figure 6 and so on: In all these figures, the text is blurred and not readable. Change all the pictures with higher resolution.

Author’s response: Dear reviewer, thank you for your comment. We revised the quality of all figures accordingly. 

  1. Introduction is little weak. Please improve and clearly specify the research gap and novelty of the paper.

Author’s response: Dear reviewer, thank you. We check the introduction and the whole manuscript making the necessary improvements.

  1. Check and improve the technical English. Break the long sentences.

Author’s response: Thank you very much for the comment, the whole manuscript revised accordingly to your suggestion.

  1. Conclusion: Provide crisp concluding points of the work.

Author’s response: Dear reviewer, we tried to implement your advice.

  1. How are the results validated?

Author’s response: Thank you for your question. The most suitable locations for the implementation of the HPHS upper reservoir are validated with the results provided from PPC. Particularly, two of the proposed sites have been recently suggested in pre-feasibility studies, which are currently in progress.

Reviewer 3 Report

A major revision is needed. I strongly recommend that the authors address the following.

1- There is no information from experts. Who are the experts in this research? How were they chosen? What is their profile (education, work experience, the field of expertise, etc.)? A method based on expert preferences has been used to achieve the results, and these are very important.

2- Avoid repetitions. I can see several repetitions at different places in this paper. Thorough proofreading is required.

3- The consistency rate (CR) in pairwise comparisons should be calculated. If the consistency rate is not appropriate, the research results are not acceptable.

4-  There is no linguistic term information to compare criteria. It should be added in the form of a table or text along with their corresponding numbers.

5-  The positive or negative type of criteria should be specified. Also, it is recommended to separate factors, criteria, and constraints well.

6- Hierarchical structure diagram including criteria and alternatives should be formed. This is the first step in the AHP method.

7- The sum of ui for each row is not shown and only the final weights are visible.

8- The data of each of the alternatives must be specified (for example, the numerical values of the distance of each alternative from the river in (m) or provide a range clearly for alternatives: >xxx <xxx). The data are not satisfactory to me. It is suggested to provide supplementary files.

9- The quality of all figures throughout the paper is very low. Considering that the results are shown in the figures, the authors should fundamentally improve the quality of the figures. If case of unable to make the figures clear, suggest converting them into one table.

10- The table 3 caption must be reworded. ranking the weights is vague. the weight provides ranking itself. Likewise in lines 534-535.

11- Managerial implications, future research suggestions, and research limitations should be considered.

Author Response

We appreciate the opportunity to submit the revised version of the paper " GIS-based assessment of Hybrid Pumped Hydro Storage as a potential solution for the clean energy transition: The Case of Kardia Lignite Mine, Western Greece" for consideration by the Sensors Journal. We value the resources and time you invested in offering comments on our manuscript. We are also appreciative of the helpful comments and enhancements made to our paper. Please see below, for a detailed response to the reviewers’ comments and concerns.

Reviewer 3: Major revision

  1. There is no information from experts. Who are the experts in this research? How were they chosen? What is their profile (education, work experience, the field of expertise, etc.)? A method based on expert preferences has been used to achieve the results, and these are very important.

Author’s response:  Dear reviewer, thank you for the valuable comment. The provided methodology and selected criteria were based on the literature, and on the judgement of industrial experts of PPC, with expertise in coal mines and specifically in the field of mining and geoengineering, geology and hydrogeology, industrial safety, environmental engineering and sustainable energy technologies. Additionally, the whole process was implemented in the context of ATLANTIS (RFCS) European research program.

  1. Avoid repetitions. I can see several repetitions at different places in this paper. Thorough proofreading is required.

Author’s response: Indeed, there were repetitions in the manuscript and we tried to make all the necessary modifications.

  1. The consistency rate (CR) in pairwise comparisons should be calculated. If the consistency rate is not appropriate, the research results are not acceptable.

Author’s response: Dear reviewer, thank you for pointing this out. Indeed, the CR is a mandatory calculation that needs to be presented. Particularly, the CR value is 0.07 and is presented in the following screenshot. In addition, we have included it in the revised manuscript.

  1. There is no linguistic term information to compare criteria. It should be added in the form of a table or text along with their corresponding numbers.

Author’s response: Thank you for pointing this out. We have included it, in tabular format as Table 3.

  1. The positive or negative type of criteria should be specified. Also, it is recommended to separate factors, criteria, and constraints well.

Author’s response: Dear reviewer, thank you for highlighting this information. We made the necessary changes in the whole manuscript. The criteria were also categorised according to their impact (positive or negative) based on the proximity analyses and AHP ranking.

  1. Hierarchical structure diagram including criteria and alternatives should be formed. This is the first step in the AHP method.

Author’s response: Dear reviewer, thank you for your suggestion. We have created a diagram the includes the criteria and alternatives.

  1. The sum of ui for each row is not shown and only the final weights are visible.

Author’s response: Thank you for your comment. Specifically, for the calculation of the weights, a Normalised matrix is presented below. Particularly, the weights were calculated by utilising the geometric mean of each line (ui) and dividing it by the sum of the geometric mean of all rows (uk) of the matrixes. Regarding the sum of ui is presented in the manuscript.

  1. The data of each of the alternatives must be specified (for example, the numerical values of the distance of each alternative from the river in (m) or provide a range clearly for alternatives: >xxx <xxx). The data are not satisfactory to me. It is suggested to provide supplementary files.

Author’s response: Dear reviewer, thank you for your valuable comment. All the criteria were classified into five classes using the Natural Breaks (Jenks) method. We have included a new table in the revised text that illustrates the range values of each class for the selected criteria.

  1. The quality of all figures throughout the paper is very low. Considering that the results are shown in the figures, the authors should fundamentally improve the quality of the figures. If case of unable to make the figures clear, suggest converting them into one table.

Author’s response: Dear reviewer, thank you for your comment. We revised the quality of all figures accordingly. 

  1. The table 3 caption must be reworded. ranking the weights is vague. the weight provides ranking itself. Likewise in lines 534-535.

Author’s response: Revised accordingly. 

  1. Managerial implications, future research suggestions, and research limitations should be considered.

Author’s response: Dear reviewer, we have integrated the whole manuscript in order to optimized it according to your comments.

Reviewer 4 Report

It is an interesting paper, but it was extremely difficult to analyze it, because figures are produced in a very poor resolution. I could barely read what was written in the figures - all of then. So, to better understanding, the figures must be improved.

Author Response

We are grateful for the opportunity to submit the revised version of the paper " GIS-based assessment of Hybrid Pumped Hydro Storage as a potential solution for the clean energy transition: The Case of Kardia Lignite Mine, Western Greece" for consideration by the Sensors Journal. We are happy to know that you are satisfied with our manuscript. We are also appreciative for the time you have spented in offering helpful comments to improve our paper. For a detailed response to the reviewers' comments and concerns, please see the section below.

Reviewer 4:

  1. Figures are produced in a very poor resolution.

Author’s response: Dear reviewer, thank you for your comment. We revised the quality of all figures accordingly. 

Round 2

Reviewer 2 Report

Accept

Reviewer 3 Report

I would like to thank the authors for their careful editing. The authors have addressed the comments and the quality of the paper has improved very well.